# Early Childhood Science Education from 0 to 6: A Literature Review

**Gillian O'Connor *** , **Glykeria Fragkiadaki, Marilyn Fleer and Prabhat Rai**

Faculty of Education, Monash University, Frankston 3199, Australia; glykeria.fragkiadaki@monash.edu (G.F.); marilyn.fleer@monash.edu (M.F.); prabhat.rai@monash.edu (P.R.)
\* Correspondence: gillian.oconnor1@Monash.edu

**Abstract:** Over the past three decades, our understanding of science learning in early childhood has improved exponentially and today we have a strong empirically based understanding of science experiences for children aged three to six years. However, our understanding of science learning as it occurs for children from birth to three years, is limited. We do not know enough about how scientific thinking develops across the first years of life. Identifying what we do know about science experiences for our youngest learners within the birth to three period specifically, is critical. This paper reviews the literature, and for the first time includes children in the birth to three period. The results are contextualised through a broader review of early childhood science education for children aged from birth to six years. Findings illustrated that the empirical research on science concept formation in the early years, has focused primarily, on children aged three to six years. The tendency of research to examine the *process* of concept formation in the birth to three period is also highlighted. A lack of empirical understanding of science concept formation in children from birth to three is evident. The eminent need for research in science in infancy–toddlerhood is highlighted.

**Keywords:** science; early childhood; science education; literature review; concepts; infants; toddlers; preschoolers





## 1. Introduction

We are living in times characterised by an explosion of scientific knowledge [1] and rapid rates of innovation in technology [2]. Globally, however, there exists a decreasing trend in student interest in science upon school completion [3] and in Australia, students' performance in science is declining [4]. It is now widely accepted that early science learning experiences are essential for the development of children's scientific knowledge and inquiry skills [5]. Appropriate scientific work can and should begin as early as possible for all children [5–7]. Concerningly, however, research has shown that current early years provisions fail to meet children's potential [8]; young children's science learning is not being systematically stimulated [9] and there are significantly fewer opportunities for young children to engage with science activities in comparison to other content areas [10]. In addition, many early childhood teachers feel discomfort when teaching sciences [11], and have expressed concern at the lack of appropriate pedagogical strategies [12,13]. The eminent need to provide more quality and challenging science experiences in early childhood is highlighted [8].

Reflecting increasing recognition of the ability of young children to engage in science learning [14–16] and the paramount significance of early science learning to later science outcomes [17], early childhood science education research as a distinct research field, has increased significantly over the past three decades [18]. Within the field, however, studies have focused predominantly on children aged three to six years with only a limited pool addressing science learning in the birth to three period specifically. Consequently, we have a strong, empirically based understanding of science experiences for children aged three

to six years. In contrast, however, our understanding of science learning in infancy and toddlerhood, as it occurs for children from birth to three years, is limited.

Seeking to develop our understanding of a crucial yet largely unknown area of science education research, the current study aimed to review the empirical literature on science concept formation in the birth to three period (within the context of the wider early childhood science education literature). An empirical literature search was conducted over the period March 2020 to January 2021. Peer reviewed journal articles examining science concept formation in pre-school settings (birth to six years), from 1990 to date, were included. Findings illustrated that the empirical research on science concept formation in early childhood, has focused primarily on children aged three to six years (50 of 57 studies identified). A lack of empirical understanding of science concept formation as it occurs from birth to three was found.

This paper presents the findings of the literature review. An overview of the methodological framework used to conduct the literature review is firstly presented and the categorisation process used to discuss the studies identified, is outlined. Overall findings of the review are then depicted graphically (Figure 1) with individual characteristics of the included studies presented in Tables 3–5. Findings of the review are then discussed in accordance with the identified categories (a detailed explanation of which is later provided). For each category, a brief summary of the literature on children aged three to six years is firstly provided. This is then followed by a more in-depth discussion of the literature on children from birth to three years.

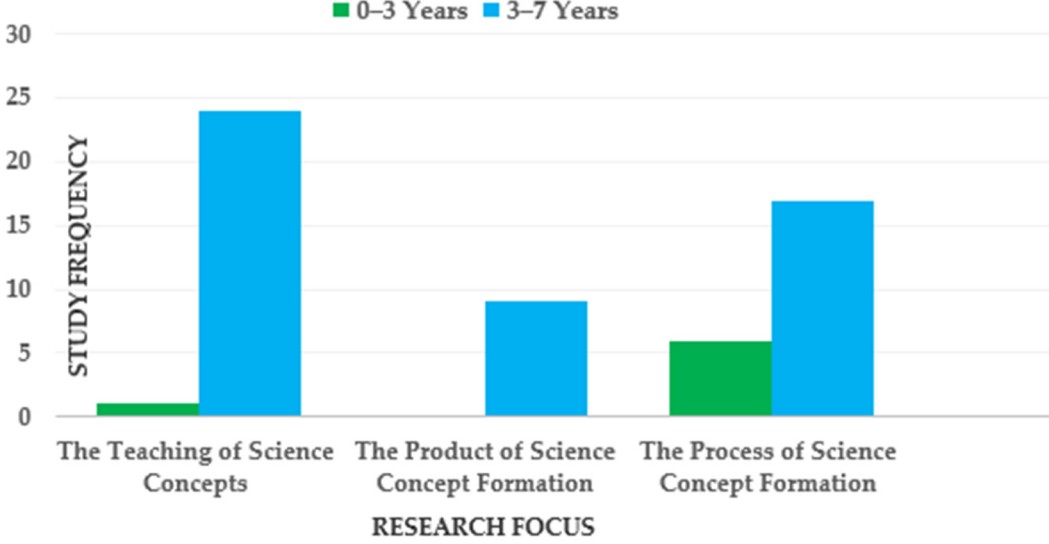

**Figure 1.** The Empirical Literature on Science Concept Formation in Early Childhood (Peer Reviewed International Journal Articles From 1990 to Date) by Research Focus and Participant Age.

## 2. Methodological Framework of the Literature Review

Reviewing to our knowledge, for the first time the empirical literature on science concept formation in the birth to three period specifically, this paper provides critical insight into our current understanding of science experiences of infants and toddlers. Gaps that exist in the literature regarding infant and toddlers' thinking in science are highlighted and the eminent need for research emphasised. The following subsections describe the methodological framework used to conduct the review.

### 2.1. Methods

An empirical literature search was conducted over the period March 2020 to January 2021 using the following databases: A+ education, ERIC, ProQuest education journals, PsycINFO and SCOPUS. Peer reviewed journal articles (written in English), examining

science concept formation in the early years setting from 1990 to date, were included. Only studies examining science concept formation in pre-school age children were included. This differed in accordance with the age of formal school commencement across countries. Studies examining science concept formation within the broader context of Science, Technology, Engineering and Mathematics (STEM) or Science, Technology, Engineering, Art and Mathematics (STEAM), were excluded. Studies in which the overall research focus did not relate to concept formation and/or children's thinking in science, were also excluded. The inclusion and exclusion criteria are presented in Table 1.

**Table 1.** Literature search inclusion and exclusion criteria.

| Category | Inclusion Criteria | Exclusion Criteria |
| --- | --- | --- |
| Research Aim/Key Words | Studies focusing on science concept formation/children's "thinking" in science | Studies in which science *concept formation* was not of primary interest |
| Topic | Science | STEM/STEAM |
| Age | Pre-school age children (country dependent) | Children in formal schooling |
| Area | International | None |
| Time | 1990 to date | Prior to 1990 |
| Type | Peer reviewed journal articles | Books/book chapters/non-academic articles, editorials, conference proceedings |
| Language | English | None-English |

Table 1 provides an understanding of the way in which studies were identified for inclusion in the literature review. The next subsection outlines the way in which the studies included, were categorised for the purpose of analysis.

*2.2. Categorisation*

The studies identified in the literature search were categorised into three groups according to the overall research focus; The Teaching of Science Concepts (Pedagogical Practices) (Category 1), The Product of Science Concept Formation (Conceptual Understandings/Demonstrated Capabilities) (Category 2) and The Process of Science Concept Formation (Development Over Time) (Category 3). A description of the categories and further sub-categorisation is now provided.

2.2.1. Category 1: The Teaching of Science Concepts (Pedagogical Practices)

Studies examining science concept formation in relation to pedagogical practices were grouped together and categorised; Category 1: The Teaching of Science Concepts (Pedagogical Practices). Studies were then further grouped into 2 sub-categories according to the more specific, research aim; studies exploring the effectiveness of specific teaching interventions/educational programs in relation to science concept formation (1A); studies exploring the effect of individual differences amongst educators in relation to science concept formation (1B).

2.2.2. Category 2: The Product of Science Concept Formation (Conceptual Understandings/Demonstrated Capabilities)

Studies that explored children's individual conceptual science understandings were grouped together and categorised; Category 2: The Product of Science Concept Formation (Conceptual Understandings/Demonstrated Capabilities). Studies were then further grouped into 2 sub-categories according to the more specific, research aim; studies exploring pre-school age children's conceptual understandings of science concepts (2A); studies exploring the age (biologically) that children begin developing scientific reasoning skills (2B).

2.2.3. Category 3: The Process of Science Concept Formation (Development over Time)

Studies that focused on examining the process of concept formation were grouped together and categorised; Category 3: The Process of Science Concept Formation (Development over Time). Studies were then further grouped into 2 sub-categories according to the more specific, research aim; studies seeking to explore and understand *how* children are developing their understandings of science concepts (3A); studies seeking to explore and understand the role the teacher plays in creating conditions for children developing conceptual understandings in science (3B).

Studies in Group 3A examine the *process* of science concept formation (how children are developing an understanding of science concepts over time). These studies differ to studies in Group 2A where the focus of research is on identifying children's conceptual understandings and/or ability to engage in science learning at a specific point in time.

Studies in Group 3B explore the way in which elements of the teachers' role influence the process of concept formation (how early childhood teachers create the conditions for the formation of science concepts). These studies therefore differ to studies in Category 1 (The Teaching of Science Concepts) where the focus of research is on examining the efficacy of specific teaching methods, interventions or instructional strategies in promoting scientific understanding.

Table 2 provides an overview of the categorisation. The frequency of papers by category and age of participants is also presented.

**Table 2.** Studies regarding science concept formation in early childhood published from 1990 to date by research aim and participant age.

| Research Area | Research Focus | Frequency 3 to 7 Years | Frequency 0 to 3 Years |
|---|---|---|---|
| CATEGORY 1: The Teaching of Science Concepts (Pedagogical Practices) | To explore the effectiveness of specific teaching interventions/educational programs (in relation to science concept formation) (1A). | 20 | 1 |
| | To explore individual differences amongst teachers (in relation to science concept formation) (1B). | 4 | 0 |
| | Total Frequency: | 24 | 1 |
| CATEGORY 2: The Product of Science Concept Formation (Conceptual Understandings/Demonstrated Capabilities) | To explore pre-school age children's conceptual understandings of science concepts (2A) | 7 | 0 |
| | To explore the age (biologically) that children begin developing scientific reasoning skills (2B) | 2 | 0 |
| | Total Frequency: | 9 | 0 |
| CATEGORY 3: The Process of Science Concept Formation (Development Over Time) | To explore and understand *how* children are developing their understandings of science concepts (3A) | 13 | 5 |
| | To explore and understand the role the teacher plays in creating conditions for children developing conceptual understandings (3B) | 4 | 1 |
| | Total Frequency: | 17 | 6 |
| | TOTAL (overall): | 50 | 7 |

Table 2 presents the frequency of studies included in the review by categorisation according to the research focus and research aim. Table 2 shows that of the 57 studies identified for inclusion in the review, 50 related to children aged three to six years and 7 related to children in the birth to three period. Table 2 also illustrates that 25 studies examined science concept formation in relation to the teaching of science concepts (Category 1) of which 1 related to children in the birth to three period; 9 studies examined science concept formation in relation to the product of science concept formation (conceptual understandings/demonstrated capabilities) (Category 2), all of which involved children aged three to six years; 23 studies examined the process of science concept formation (Category 3), 6 of which related to children in the birth to three period.

## 3. Findings and Discussion

### 3.1. Individual Study Characteristics

A summary of the main characteristics of the studies included in the review are now presented in accordance with the categories discussed. Three different tables are presented. The first table (Table 3) summarises the main characteristics of studies in Category 1 (The Teaching of Science Concepts; Pedagogical Practices). The second table (Table 4) summarises the main characteristics of studies in Category 2 (The Product of Science Concept Formation; Conceptual Understandings/Demonstrated Capabilities). The third table (Table 5) summarises the main characteristics of studies in Category 3 (The Process of Science Concept Formation; Development over Time). Within each table, the studies are presented in alphabetical order of the author's names.

Table 3 presented the characteristics of empirical studies examining science concept formation in the early years (birth to six years) in relation to the teaching of science concepts (pedagogical practices). Table 4 now follows, presenting characteristics of empirical studies examining science concept formation in the early years (birth to six years) in relation to the product of science concept formation (conceptual understandings/demonstrated capabilities).

Table 4 presented the characteristics of empirical studies examining science concept formation in the early years (birth to six years) in relation to the product of science concept formation (conceptual understandings/demonstrated capabilities). Table 5 now follows, presenting characteristics of empirical studies examining science concept formation in the early years (birth to six years) in relation to the process of concept formation (development over time).

Table 5 presented the characteristics of empirical studies examining science concept formation in the early years (birth to six years) in relation to the process of concept formation (development over time).

Collectively, Tables 3–5 provide an overview of the general characteristics of studies which have examined science concept formation in the early years. A summary of the overall findings and ways in which science concept formation has been examined, from the studies included in the review is now provided.

### 3.2. Findings

The findings from the literature review of published empirical works on science concept formation in early childhood are illustrated through a bar graph of categories (Figure 1), followed by a discussion of what was determined through an analysis using the categories discussed; 1. The Teaching of Science Concepts (Pedagogical Practices); 2. The Product of Science Concept Formation (Conceptual Understandings/Demonstrated Capabilities), and 3. The Process of Science Concept Formation (Development over Time).

Figure 1 provides a visual representation of the empirical literature on science concept formation in early childhood by research focus and participant age. The extent with which research has involved children aged three to six years (in contrast to children in the birth to three period) is highlighted. 50 studies involved children aged three to six years, 7 studies related to children in the birth to three period. A limited empirical understanding of science experiences of children in the birth to three period (in contrast to that of older pre-school age children) is highlighted.

A discussion of the studies identified in the review is now provided by category. Having been previously reviewed the literature on children aged three to six years are firstly summarised and a more detailed discussion of studies involving children aged birth to three then given.

**Table 3.** Characteristics of empirical studies examining science concept formation in the early years (birth to six years) in relation to the teaching of science concepts (pedagogical practices).

| Research Focus | Reference | Research Aim | Science Concept(s)/Skills | Country/Sample | Methods | Findings/Conclusions |
|---|---|---|---|---|---|---|
| To explore the effectiveness of specific teaching interventions/educational programs (in relation to science concept formation) (1A). | Dejonckheere et al., 2016 [19] | Tested and integrated the effects of an inquiry-based didactic method for preschool science. | Scientific reasoning skills | Belgium, 57 children aged 4 to 6 years | Structured interviews (pre-post-test) | The inquiry-based didactic method encouraged children's spontaneous exploratory activities. |
| | Dogru and Seker, 2012 [20] | To determine the effect of science "activities" on cognitive development and science concept acquisition skills. | Astronomy | Turkey, 48 children aged 5 to 6 years | Interviews, participant drawings | "Science activities" is an effective technique in the acquisition of basic concepts related to "the Earth, Sun and Moon". |
| | Hadzigeorgiou, 2002 [21] | Investigated the efficacy of structured hands-on activities to facilitate preschool children to construct the concept of mechanical stability. | Mechanical stability | Greece, 37 children, 4.5 to 6 years | Video recordings | Appropriately structured activities involving children's action on objects, complemented with a scaffolding strategy, help children construct the concept of mechanical stability. |
| | Hannust and Kikas, 2007 [22] | Analysed the influence of instruction on the development of astronomical knowledge. | Astronomy | Estonia, 113 children aged 5 to 7 years | Video recordings, interviews, drawing tasks | Children acquired factual information easily and over-generalized new knowledge easily: materials used in teaching may promote the development of non-scientific notions. |
| | Hong and Diamond, 2012 [23] | Examined the efficacy Responsive Teaching (RT) and the combination of Responsive Teaching and Explicit Instruction (RT + EI) to facilitate children's learning of science. | Floating/sinking, scientific problem-solving skills | United States America (USA), 104 children aged 4 to 5 years | Interviews (pre-post-test) | Children learned science concepts and vocabulary better when either responsive teaching or the combination of responsive teaching and explicit instruction was used. |
| | Kallery et al., 2009 [13] | Examined the extent to which the teaching practices adopted by early-years educators are successful in supporting young children's understanding in science. | Physics, biology, astronomy | Greece, 11 teachers | Field notes | The didactical activities analysed did not promote scientific understanding. Scientific activity was mainly confined to the representational level. |
| | Kalogiannakis et al., 2018 [24] | Examined whether the "picture story reading" method can be beneficial for young children learning about magnetism. | Magnetism | Greece, 30 children aged 4 to 5.5 years | Structured interviews, children's drawings | Pictorial story reading in kindergarten, together with suitable questions by the teacher were effective in aiding understanding of magnetism. |
| | Kambouri-Danos et al., 2019 [25] | Examined the way in which the construction of a precursor model can support children's scientific learning. | Water (change of states) | Greece, 91 children aged 5 to 6 years | Interviews | It is possible for children aged 5 to 6 years, to consistently approach a complete sequence of water state changes, as part of a specifically designed teaching intervention. |

**Table 3.** *Cont.*

| Research Focus | Reference | Research Aim | Science Concept(s)/Skills | Country/Sample | Methods | Findings/Conclusions |
|---|---|---|---|---|---|---|
| | Kolokouri, and Plakitsi, 2016 [26] | Examined the connection of Cultural Historical Activity Theory with Science Education in the early grades. | Light/shadows, colour | Greece, 92 "pre-primary" children | Video recordings, interviews, field notes | The learning of scientific concepts is a creative component of methods, interactions and social practices. CHAT is a promising field for Science Education in the early grades. |
| | * Lloyd et al., 2017 [27] | Developed and delivered a programme of activities aimed at encouraging parents' confidence in their own ability to support emergent scientific thinking. | Forces, materials and their properties, the living world | England, 19 care givers, 26 children aged 0 to 5 years | Audio recordings, questionnaires | Parental interaction enhanced children's learning at least as much, if not more, than practitioner interventions. Mediation of experience by familiar adults facilitated enjoyment, encouraged natural curiosity. |
| | Nayfeld et al., 2011 [28] | Developed an intervention to increase childrens use of science materials (in preschool classrooms) during "free choice" time. | Properties of matter | USA, 84 children aged 3 to 5 years | Time sampling method, interviews | Children's voluntary presence and exploration in the science area increased after the intervention. Children demonstrated improved conceptual knowledge. |
| | Peterson, 2009 [29] | Examined the use of narrative and paradigmatic modes of explanation in large group discussions about science in preschool classrooms. | Measurement, mapping, light, properties of matter, natural habitats | USA, 29 teachers, 479 pre-school children | Video recordings | Students in the "science curriculum" classrooms were exposed to a higher frequency of paradigmatic explanations and produced a higher relative frequency of paradigmatic explanations. |
| | Ravanis, 1994 [30] | Explored learning situations that take place within the framework of a constructivist pedagogy. | Magnetism | Greece, 79 children, mean age 5.5 years | Video recordings, field notes | Children were able to discover the action of attractive magnetic forces on nonmagnetic materials, the attractive and repulsive forces between magnets, and distinguish between magnetic and nonmagnetic material. |
| | Ravanis et al., 2004 [31] | Investigated the effect of a socio-cognitive teaching strategy on young children's understanding of friction. | Friction | Greece, 68 children aged 5 to 6 years | Structured interviews (pre-post-test) | Evidence for the effect of the socio-cognitive strategy on children's understanding of a "precursor model" for the concept of friction was found. |

**Table 3.** *Cont.*

| Research Focus | Reference | Research Aim | Science Concept(s)/Skills | Country/Sample | Methods | Findings/Conclusions |
|---|---|---|---|---|---|---|
| | Ravanis and Pantidos, 2008 [32] | Explored Piagetian and Post-Piagetian strategies for children working with magnets. | Magnetism, friction | Greece, 41 children aged 5.5 to 6.5 years | Video recordings | The differing educational content of the two pieces of research led to different levels of progress in children's thought. Successful changes in children's thought occurred only in the case of magnetic properties. |
| | Ravanis et al., 2013 [33] | Investigated the effect of a socio-cognitive teaching strategy on young children's understanding of light. | Light | Greece, 170 children aged 5.5 to 6.5 years | Structured interviews (pre-post-test) | Evidence for the effect of the socio-cognitive strategy on enhancing children in constructing a "precursor model" for the concept of light was found. |
| | Strouse and Ganea, 2016 [34] | Investigated whether adult prompting during the reading of an electronic book enhanced children's understanding of a biological concept. | Electricity | USA, 91 children aged 4 years | Structured interviews (pre-post-test) | Under some circumstances, electronic prompts built into touchscreen books can be as effective at supporting conceptual development as the same prompts provided by a co-reading adult. |
| | Tenenbaum et al., 2004 [35] | Investigated the effectiveness of a combined museum and classroom intervention project on science learning in low-income children. | Water related concepts | USA, 48 kindergarten children | Interviews | In general, the program supported children's science literacy development with regard to both concept complexity and content knowledge. |
| | Valanides et al., 2000 [36] | Investigated the effectiveness of a teaching intervention designed to teach pre-school age children astronomical concepts | Astronomical concepts | Greece, 33 children aged 5 to 6 years | Interviews | The majority of children accepted that the Sun and the Earth are separate spherical objects, but fewer children attributed the day/night cycle to rotation of the Earth on its axis. |
| | Walan and Enochsson, 2019 [37] | Explored the outcome of using a model in which drama and storytelling were connected to facilitate learning processes in science for young children. | Human biology (the immune system) | Sweden, 25 children aged 4 to 8 years | Semi structured interviews, drawings | The combination of storytelling and drama as an instructional strategy has a positive potential when it comes to teaching children science. |
| | Zacharia et al., 2012 [38] | Investigated whether physicality (actual and active touch of concrete material) is a necessity for science experimentation learning at the kindergarten level. | Balance | Cyprus, 80 children aged 5 years | Structured interviews (pre-post-test) | Physicality appears to be a prerequisite for students' understanding of concepts (concerning the use of a beam balance), only when the students have incorrect prior knowledge of what a beam balance does. |

**Table 3.** *Cont.*

| Research Focus | Reference | Research Aim | Science Concept(s)/Skills | Country/Sample | Methods | Findings/Conclusions |
|---|---|---|---|---|---|---|
| To explore individual differences amongst teachers (in relation to science concept formation) (1B). | Fleer, 2009a [39] | Examined teacher philosophy and pedagogical practices within the context of an analysis of children's concept formation within early childhood settings. | Water related concepts, properties of matter | Australia, 2 teachers, 24 children aged 4 to 5 years | Interviews, video recordings, photographic documentation | Teacher philosophy about how young children learn is a significant contributing factor to learning in science. |
| | Gerde et al., 2018 [40] | Investigated the nature of teachers' domain-specific self-efficacy. | Common science activities (early childhood science curricula) | USA, 67 teachers | Teacher surveys | Domain-specific self-efficacy was lower for science than literacy. Self-efficacy for science, related to teacher's frequency of engaging children in science instruction. |
| | Fleer et al., 2014 [41] | Examined how the environment is perceived by teachers for creating opportunities for science learning. | Common science activities (early childhood science curricula) | Australia, 65 children, 3.3 to 4.6 years | Video recordings, photographs | A "sciencing attitude" on the part of the teacher affords meaningful science learning for preschool children. |
| | Gomes and Fleer, 2018 [42] | Examined how teachers use the preschool environment to promote the teaching of science concepts. | Experiment based science activities | Australia, 2 pre-school teachers | Video recordings | Teachers in the same preschool setting have different levels of science awareness for the possibilities of informally teaching science. |

* Studies examining science concept formation in the birth to three period.

**Table 4.** Characteristics of empirical studies examining science concept formation in the early years (birth to six years) in relation to the product of science concept formation (conceptual understandings/demonstrated capabilities).

| Research Focus | Reference | Research Aim | Science Concept(s)/Skills | Country/Sample | Methods | Findings/Conclusions |
|---|---|---|---|---|---|---|
| To explore pre-school age children's conceptual understandings of science concepts (2A). | Akerson et al., 2011 [43] | Examined the capability of young children to learn about the Nature of Science (NOS). | NOS | USA, 18 children; kindergarten to 2nd grade. | Structured interviews | Children improved their understandings of NOS in each setting. Kindergarten children are developmentally capable of conceptualizing NOS when it is taught to them. |
| | Allen, 2017 [44] | Explored aspects of pre-schoolers' ecological understandings. | Ecology | England, 70 children aged 3 to 5 years | Structured interviews | 5-year-old children arecapable of grasping concepts inherent in food chain topics scheduled to appear later in their schooling. |
| | Borgeding and Raven, 2018 [45] | Investigated pre-schoolers understandings of fossils in the context of a week-long informal science camp. | Fossils | USA, 15 children aged 3 to 6 years | Structured interviews | Clear age and object-related trends for living/non-living distinctions, teleological reasoning, origins, and object ages were noted. |

**Table 4.** *Cont.*

| Research Focus | Reference | Research Aim | Science Conept(s)/Skills | Country/ Sample | Methods | Findings/Conclusions |
|---|---|---|---|---|---|---|
| | Constantinou et al., 2013 [46] | Examined the ability of young children to construct operational definitions in magnetism and the importance of scaffolding the learning environment. | Magnetism | Cyprus 165 children aged 4 to 6 years | Structured Interviews | Cognitive maturation is not the main determinant factor that shapes the performance pattern of these children. The necessary role of scaffolding the curriculum materials to achieve successful learning is highlighted. |
| | Forman, 2010 [47] | Explored the relations between young children's play and scientific thinking. | Push/pull, force, scientific thinking skills | USA 3 children aged 3 to 4 years | Video recordings | The small experiments, inventions, strategies, and pauses in young children's play reveal a legitimate form of scientific thinking. Science and play represent a frame of mind. |
| | Krnel et al., 2005 [48] | Explored the development of the concept of matter. | Matter | Slovenia, 84 children aged 3 to 13 years | Structured interviews | Young children (age 3 to 7 years) experience objects and substances by acting upon them or using them. |
| | Smolleck and Hershberger, 2011 [49] | Investigated the conceptions and misconceptions of young children related to science concepts, skills, and phenomena. | Matter, magnetism, density | USA, 63 children aged 3 to 8 years | Video recordings | Findings reveal the most common conceptions related to matter, magnetism, density, and air. |
| | Solomonidou and Kakana, 2000 [50] | Investigated the representations and primary notions children create, on the basis of their everyday experience, for common electrical devices and electric current. | Electricity | Greece, 38 children aged 5.5 to 6.5 years | Semi structured interviews | Children had no difficulty in recognising and naming the electric appliances they were familiar with. Children held a variety of preconceptions about electric current. |
| To explore the age (biologically) that children begin developing scientific reasoning skills (2B). | Piekny and Maehler, 2013 [51] | Investigated when scientific reasoning skills emerge and whether these abilities develop synchronously during childhood. | Scientific reasoning skills | Germany, 223 children, 4 to 13.5 years | Structured interviews | The three cognitive components of domain general scientific reasoning emerge asynchronously during early and middle childhood. |
| | Piekny et al., 2014 [52] | Investigated how and when children's ability to evaluate evidence and their understanding of experimentation develops, (between ages of 4 and 6). | Scientific reasoning skills | Germany, 138 children,4 to 6 years | Structured interviews | The ability to evaluate evidence is well developed at age four and increases steadily and significantly over time. Children's understanding of experimentation increases significantly between the ages of 5 and 6. |

**Table 5.** Characteristics of Empirical Studies Examining Science Concept Formation in The Early Years (Birth to Six Years) in Relation to The Process of Concept Formation (development over time).

| Research Focus | Reference | Research Aim | Science Concept(s)/Skills | Country/ Sample | Methods | Findings/Conclusions |
|---|---|---|---|---|---|---|
| To explore and understand *how* children are developing their understandings of science concepts (3A). | Christidou, and Hatzinikita, 2006 [53] | Explored the different types and characteristics of preschool children's explanations of plant growth and rain formation. | Natural phenomena | Greece, 60 children aged 4.5 to 6.5 years | Semi-structured interviews | Children are relatively selective in regard to the explanatory type they use when discussing natural phenomena. Naturalistic explanations have different characteristics according to the phenomenon under discussion. |
| | Fleer, 2009b [54] | Examined the reciprocity between everyday thinking and scientific thinking during playful encounters in early childhood centres. | Physical attributes of materials | Australia, 48 children 4 to 5 years (4 focus children) | Video recordings, photographs, interviews | Playful events provide an important conceptual space for the realisation of dialectical relations between everyday concepts and science concepts. The "teacher as mediator" is central. |
| | Fleer, 2013 [55] | Examined the emotional nature of scientific learning; affective imagination in early childhood science learning. | Heating/cooling, light | Australia, 53 children aged 3 to 4 years | Video recordings | Identify 5 key elements that draw attention to the relations between emotions and cognition in science learning. |
| | Fleer, 2019 [56] | Examined how imaginative play promotes scientific learning and how teachers engaged children in scientific play. | Microbes and Microscopic organisms | Australia, 3 pre-school teachers, 26 children aged 3.6 to 5.9 years | Video observations, photographs, interviews | The building of collective scientific narratives alongside of discourses of wondering were key determinants of science learning in play-based settings. The Scientific Playworlds is a possible model for teaching science in play-based settings. |
| | Fragkiadaki and Ravanis, 2014 [57] | Explore the dynamic of pre-schoolers' interactions during the approach of basic science concepts. | Natural Phenomena | Greece, 16 children aged 4 to 6 years | Open type, semi-structured conversations | Different types of substantial interactions between the children couples were identified. Through a "conversational approach", organized in couples, we can foster and enhance science thinking and learning in early childhood. |
| | Fragkiadaki and Ravanis, 2016 [58] | To structure a cultural-historical understanding on how early childhood children experience science and how they develop scientific thinking as they interact with the social, cultural and material world. | Natural phenomena | Greece, 1 child aged 5.2 years | Expanded, open-type conversations | Insights into how a certain social situation between children and educators in kindergarten settings becomes the unique social situation of a child's development was gained. |

**Table 5.** *Cont.*

| Research Focus | Reference | Research Aim | Science Concept(s)/Skills | Country/ Sample | Methods | Findings/Conclusions |
|---|---|---|---|---|---|---|
| | Fragkiadaki and Ravanis, 2015 [59] | Examined children's representations on the phenomena of the natural world and on natural science concepts. | Natural phenomena | Greece, 16 children aged 4.5 to 6 years | Expanded, open type conversations between children and researchers | Children use different types of representations dominated by the nature of the substance under study. Children possess a range of ideas and explanatory mechanisms regarding the natural phenomenon and they are able to reason about them. |
| | Fragkiadaki et al., 2019 [60] | Aimed to provide a cultural-historical understanding on how children form relevant representations of clouds as well as how children's understandings are transformed and developed through communications with others. | Natural phenomena | Greece, 16 children aged 4.5 to 6 years | Expanded, open-type conversations | When children construct everyday understandings of natural phenomenon, they draw upon multiple discussions, collaborations, social experiences, knowledge, practices, values, attitudes, tools, signs, objects, sketches, and gestures. Imagination is an important dimension of children's thinking. |
| | Fragkiadaki et al., 2021 [61] | Seek to capture and explore the dialectic interrelations between intellect, affect, and action during science experiences within early childhood educational settings. | Natural phenomena | Greece, 113 children aged 4.5 to 6.5 years | Video recordings (semi-structured conversations) | The findings made visible the processes through which children make sense and shape their understandings of the natural phenomenon during everyday educational reality. |
| | Fredj, 2019 [62] | Explored how science is done in collaborative interactions when children discuss reasons for animal diversity. | Animal biology | Sweden, 27 children aged 6 years | Video recordings | While engaged in highly collaborative interactions, the children use observations to evaluate, challenge and question each other. The character of the collaborative interactions is an important factor for how acts of doing science are carried out. |
| | * Klaar and Öhman, 2012 [63] | Explored how infants form science concepts through their actions in nature. | Natural phenomena | Sweden, 1 child aged 22 months | Video recordings Practical Epistemological Analysis (PEA) | Bodily experiences, (physical encounters) are fundamental for children's further learning about natural phenomena and processes. A methodology based on the principles of PEA allows for analyses of non-verbal, bodily actions in order to investigate toddler's physical nature experiences. |

**Table 5.** *Cont.*

| Research Focus | Reference | Research Aim | Science Concept(s)/Skills | Country/ Sample | Methods | Findings/Conclusions |
|---|---|---|---|---|---|---|
| | Larsson, 2013a [64] | To gain knowledge about what aspects of, and in what way, contextual and conceptual intersubjectivity contribute to emergent science knowledge about sound. | Sound | Sweden, 10 children aged 3–6 years | Video recordings, teacher transcripts | Emergent science knowledge is developed when it is enhanced by teachers' double move between conceptual and contextual intersubjectivity. The use of contextual and conceptual intersubjectivity contribute to bridging children's everyday understandings to scientific concepts. |
| | * Larsson, 2013b [65] | Explored preschool children's opportunities for learning about friction. | Friction | Sweden, 4 children aged 2.1 to 5.6 years | Video recordings; "shadowing" | Children are in contact with the phenomenon of friction during their play. Everyday play situations can be used by teachers to become more knowledgeable about children's understandings of the friction and direct their attention to it. |
| | * Sikder, 2015 [66] | Examined how sciene concept formation becomes part of the infants-toddlers lived everyday experience at home. | Force, water properties, heating/cooling. | Australia and Singapore 4 children aged 10 to 36 months | Video recordings, interviews | Children are learning concepts (or/and small science concepts) through the purposeful actions of parents. Possibilities of science concept formation at the infant-toddler age are not any extra effort for parents. |
| | * Sikder and Fleer, 2015 [67] | Examined social interactions in everyday family life that supports the development of science concepts for infants and toddlers. | Everyday science activities (family home) | Australia and Singapore 4 children aged 10 to 36 months | Video recordings, interviews | "Small science" can help explain how the everyday experiences of young children lay the foundation for the development of concrete "scientific" concepts. |
| | * Sikder and Fleer, 2018 [68] | Examined infant-toddler's development of science concept formation within the family context. | Everyday science activities (family home) | Australia and Singapore 4 children aged 10 to 36 months | Video recordings, interviews | Small science concepts can be developed through a special form of narrative collaboration, where parents and infants consciously consider the environment from a scientific perspective. |
| | Siry et al., 2012 [69] | Explore the nature of science learning as a social phenomenon that is discursively bound. | Water related concepts | Luxembourg, 29 children aged 4 to 6 years | Video recordings, children's photographs and paintings | By positioning scientific inquiry as a fluid process children were able to enact science collaboratively and through multimodal means. |

**Table 5.** *Cont.*

| Research Focus | Reference | Research Aim | Science Concept(s)/Skills | Country/ Sample | Methods | Findings/Conclusions |
|---|---|---|---|---|---|---|
| | Siry and Max, 2013 [70] | Examines how children enact developing understandings in science through multiple interactions. | Water related concepts | Luxembourg, 26 children aged 4 to 6 years, 1 teacher | Video recordings (researcher), video recordings and photography (children) | Children's investigations were mediated by their own speculations and explanations. Emphasis is placed on the value of students being positioned as co-constructors of science curricula. |
| | Abdo and Vidal Carulla, 2020 [71] | Explore emergent understanding of preschool-aged children about the scientific concept of "small", as used in theoretical chemistry. | Chemistry concepts | Sweden, 4 children aged 3 to 5 years | Video recordings | A process of "sustained shared thinking" could describe the teaching/learning processes evident in the children's and teacher's conversations. Sustained and shared conversations between children and teachers should stem from children's everyday experiences. |
| To explore and understand the role the educator plays in creating conditions for children developing conceptual understandings (3B). | *Fragkiadaki et al., 2020 [72] | Examined how infants in play-based settings, develop scientific understandings about their everyday world. | Sound | Australia, 13 children aged 5 months to 2 years and 3 months | Video recordings | Key elements of the teacher's pedagogical positioning were suggestive of the way through which the "ideal form" of concept formation can be introduced and supported in the infants' environment. 4 key elements for introducing science concepts in infants' everyday educational reality are proposed. |
| | Fridberg et al., 2019 [73] | Aimed to develop knowledge about the communication established between teacher and children in relation to an object of learning (intersubjective communication). | Chemistry and physics concepts | Sweden, 5 children aged 3 to 5 years, 5 teachers | Video recordings | Intersubjectivity can occur in different ways with different consequences for children's opportunities to experience the intended object of learning. |
| | Havu-Nuutinen, 2005 [74] | Examined young children's conceptual change process in floating and sinking from a social constructivist perspective. | Floating/sinking | Finland, 10 children aged 6 years | Interviews (pre-post-test) | The child's awareness and interest were raised. Through challenging the child's thinking and encouraging the flow of ideas the foundations for later scientific understanding can be developed. |
| | Pramling and Pramling Samuelsson, 2001 [75] | Explore the verbal interaction between a child and teacher focusing on how the interaction enables the child to test and prove a self formulated hypothesis. | Floating/sinking | Sweden, 1 child aged 3.3 years | Video recordings | Conceptually orientated teacher–child interactions seemed to support the children's cognitive progress in cognitive skills and guided the children to consider the reasons for flotation. |

* Studies examining science concept formation in the birth to three period.

3.2.1. Category 1. The Teaching of Science Concepts (Pedagogical Practices)

At the time of review, 25 of the 57 studies identified, examined science concept formation in the context of its relation to the teaching of science concepts (pedagogical practices); 24 related to children aged three to six years and 1 to children in the birth to three period.

The Teaching of Science Concepts (Pedagogical Practices): Three to Six Years

As earlier discussed, studies examining science concept formation in relation to pedagogical practices were further grouped into two sub-categories according to the more specific, research aim; studies exploring the effectiveness of specific teaching interventions/educational programs in relation to science concept formation (1A); studies exploring the effect of individual differences amongst teachers in relation to science concept formation (1B). Findings are now discussed in accordance with these sub-categories; 1A: Teaching Interventions/Programs; 1B: Individual Differences across Early Years Teachers.

1A.   The Teaching of Science Concepts (Pedagogical Practices) (three to six years): Teaching Interventions/Programs

The majority of studies examining science concept formation in relation to pedagogical practices, explored the extent with which specific teaching methods and/or instructional strategies commonly used within the early childhood classroom/centre, facilitated children's learning of science. Teaching approaches examined included the combination of storytelling and drama [37], pictorial story reading [24], the use of electronic prompts built into touchscreen books [34] and the necessity of "physicality" (actual and active touch of concrete material) as a pre-requisite for science experimentation [38]. A cluster of studies have examined the relation between science concept formation and more broader teaching approaches including the use of responsive teaching and explicit instruction [23], narrative and paradigmatic modes of explanation [29] and inquiry based didactic methods [19]. The relation between teaching approaches differing theoretically, and science learning has also been explored including the effect of a socio-cognitive strategy on enhancing children in constructing a "precursor model" for the concept of friction [31] and light [33], Piagetian strategies [32], Cultural Historical Activity Theory [26] and a constructivist pedagogy [30]. Providing insight into the effects of different instructional strategies on science learning in early childhood specifically, the studies identified are pivotal in assisting early childhood teachers to develop appropriate strategies for teaching sciences [37].

Empirical research examining the efficacy of teaching interventions/educational programs is fundamental in providing a strong evidence base upon which emerging early childhood science practices and curricular reforms can be built [38]. A small group of studies identified in the current review examined the efficacy of specific teaching interventions/programs, in promoting scientific understanding, within the early childhood setting. Studies identified examined interventions in relation to the development of conceptual understandings of astronomical concepts [20,36], the construction of "pre-cursor" models to support scientific learning [25], the implementation of interventions designed to increase children's voluntary exploration of science centres during free choice play [28] and the combining of a museum and classroom intervention project on science learning in low-income children [35]. Positive outcomes in relation to the development of science concepts (in children aged 3–6 years) across all the identified studies was noted. That only 4 studies identified, examined the efficacy of specific teaching interventions is perhaps reflective of concerns within the wider literature, where a lack of outcome studies to support the implementation of innovative early childhood science curricular has been highlighted [76]. The provision of a strong evidence base is fundamental to the delivery of effective programs and best practice. The need for empirical work examining the efficacy of novel curricular and innovative classroom practices for science in early childhood is highlighted.

1B.   The Teaching of Science Concepts (Pedagogical Practices) (three to six years): Individual Differences Across Early Years Teachers

A group of studies identified explored individual differences amongst teachers in relation to children's science learning experiences. Fleer [39] found individual teacher "philosophy" about how young children learn, to be a significant contributing factor to learning in science. Similarly, the level of individual teacher science "awareness" in relation to opportunities available in the environment, was found to be a contributing factor to science learning [42]. Domain specific self-efficacy has also been explored with an association found between teacher self-efficacy for science and the frequency of which children are engaged in science instruction [40].

Highlighting ways in which science opportunities provided in early childhood classrooms/centres can be enhanced (at the level of teacher education and the context of the everyday classroom) the studies on individual differences amongst teachers, have significant implications for teaching practices and child outcomes [40]. For instance, the association found between a lack of self-efficacy and time children are engaged in science instruction [40], supports the need for pre-service and in-service education programs to provide teachers with content and practices for science rather than focusing exclusively on literacy [40]. However, Fleer [39] draws caution to research in which a lack confidence and competence in teaching science is emphasised, suggesting that there is a tendency of such research to "blame the victim" whilst providing little analysis or insight into the reason. Instead, Fleer [39] argues that emphasis should be placed on individual teacher philosophy as this was found to have more of a difference to children's scientific learning than both teacher confidence in science teaching or science knowledge.

The Teaching of Science Concepts (Pedagogical Practices): Birth to Three

The literature review identified 1 study [27] in which science concept formation was examined in relation to pedagogical practices, in children aged birth to three years. This study is now discussed.

Research conducted by Lloyd et al. [27] represents the only study to empirically examine the teaching of science concepts (pedagogical practices) in relation to science concept formation of children in the birth to three period specifically. Seeking to bridge science learning across institutional practices of home and day care, Lloyd et al. [27] conducted an exploratory study of a "stay and play" service. A programme of activities aimed at encouraging parents' confidence in their own ability was delivered to support emergent scientific thinking. Findings demonstrated that the program generated children's engagement and interest and in addition, enhanced parental and practitioner confidence in their ability to promote young children's natural curiosity at home and in early childhood provision. The study found that parental interaction enhanced the children's learning at least as much, if not more than practitioner interventions. Based on this finding Lloyd et al. [27] highlight the significance of "familiar" adults in mediating young children's enjoyment and encouraging natural curiosity.

The significant role of an adult in mediating science learning experiences in infancy specifically, has been highlighted both in the context of the family home [66–68] and the educational setting [72]. As a key asset of a rich science learning environment in infancy [72], the significant role of an adult in mediating science learning experience in infancy specifically, illustrates a crucial difference in the science learning of infants in comparison to older pre-school age children. To Fragkiadaki et al. [72] (p. 19), "*the introduction of scientific concepts in the infant environment differs from a quick introduction or as something to be discovered without an adult guide*". That an assumption that teaching and learning practices used with older children can be used with infants is highlighted within the literature [77], underscores the imperative need for empirical research examining science learning in infancy period specifically.

### 3.2.2. Category 2: The Product of Science Concept Formation (Conceptual Understandings/Demonstrated Capabilities)

A substantial proportion of empirical research in science education (in general) has centred around identifying children's individual conceptual understandings at specific points in time. Most commonly, studies have used experimental clinical methods to elicit children's thinking in order to determine their specific conceptual understandings. At the time of review, 9 of the 57 studies identified, examined science concept formation in the context of its relation to conceptual understandings; all studies related to children aged three to six years.

### The Product of Concept Formation (Conceptual Understandings/Demonstrated Capabilities): Three to Six Years

Exploration of children's conceptual understandings in relation to specific concepts identified in the current review included nature of science [43], ecological understandings [44], fossils [45], and electricity [50]. Studies have also examined children's conceptions and misconceptions [49] their ability to construct operational definitions in magnetism [46] and how and when scientific reasoning skills emerge in the early childhood [51,52]. Collectively this area of research provides us with a strong empirical insight into the conceptual understandings of children aged three to six years in relation to scientific phenomena. It is, however, important to note that reliance on these studies has been cautioned in relation to the (arguably) questionable validity of study methods; many approaches used to determine how children think about science concepts have been deemed suitable to use with young children without any questioning of their reliability when applied to this age group specifically [78]. The extent with which examining science concept formation in the early childhood is associated with methodological challenges are highlighted later in the paper.

### The Product of Science Concept Formation (Conceptual Understandings/Demonstrated Capabilities): Birth to Three

No individual study examining conceptual understandings of scientific phenomena in children aged birth to three specifically was identified in the review. However, insight into a more cognitive perspective of science learning in infancy can be gained through discussion of the broader Cognitive literature. In contrast to the Piagetian view of infants as "irrational and pre-causal", a branch of cognitive research, conceptualises infants as "little scientists" or "theorists," actively attempting to build theories about the world [79]. Research centres around demonstrating the extent with which young children "think like scientists", testing hypothesis, making causal inferences and learning from informal experimentation [80]. In a 2011 review, Keil [81] concluded that young children share the cognitive "skills" of scientists, detecting correlations between seemingly unrelated phenomena, inferring causation, uncovering explanatory mechanisms and then sharing and building upon this knowledge with others. Similarly, in an overview of empirical studies in science in infancy, Gopnik [82] argues that infants learn about the world in the same way scientists do, analysing statistical data patterns, conducting experiments, and learning from the data and others. Using findings from an observational study Forman [47] argues that infants think like scientist because they use the same methods of, e.g., sensing the problem, and testing first. The small experiments, inventions, strategies, and pauses in young children's play captured in observations of toddlers' play illustrate a legitimate form of scientific thinking [47].

Painting an increasingly positive picture of young children as "active learners" [83] the (cognitive) research discussed, provides valuable insight into the cognitive processes associated with science learning in infancy. However, concerns with the cognitive premise on which the "little scientist" model is based have been highlighted. Nelson [84] (p. 6) argues that what the child brings to any conceptual domain is a built-in structure, fails to acknowledge outside influences on development; it is "*imperviousness to influences on basic cognitive processes or cognitive growth from outside the mind . . .* ". Consequently, the

complexity of cognitive development, the significance of knowledge in the social and cultural world surrounding the child, are obscured [84].

3.2.3. Category 3: The Process of Science Concept Formation (Development over Time)

In contrast to research in which the primary focus has been to identify conceptual understandings, a number of studies identified examined *how* young children develop their understandings of science concepts; the "process" of science concept formation. At the time of review, 23 of the 57 studies identified, examined the process of science concept formation; 17 studies related to children aged three to six years and 6 to children in the birth to three period.

The Process of Science Concept Formation (Development over Time): Three to Six Years

Seeking to explore how young children form conceptual understandings in an everyday context, a number of studies have explored the collaborative nature of science learning [56–58,60,62,69,70] findings of which highlight the importance of collaborative experiences. Other studies have explored the dialectic interrelations between intellect, affect, and action during science experiences [61], the emotional nature of scientific learning [55] and the reciprocity between everyday thinking and scientific thinking during playful encounters [54]. The differing ways in which children represent natural phenomena have also been explored [53,59]. The way in which elements of the teachers' role, influence the *process* of concept formation, has been explored including sustained and shared conversations between children and teachers [71], intersubjective communication (intersubjectivity) [73], contextual and conceptual intersubjectivity [64], conceptually orientated teacher–child interactions [74] and verbal interaction between a child and teacher [75].

Examining the process of science concept formation, studies in this category have primarily been conducted within a Socio-Cultural (SC)/Cultural-Historical (CH) theoretical framework. Focusing on the contexts in which children develop socio-culturally relevant activities, and the interactions with others that support and guide them [85], SC/CH research is suggested to provide a powerful and authentic approach to researching young children's thinking [6]. That as suggested by Robbins [85], individual thinking does not occur in a "vacuum" separate from activities in which people engage, highlights the value of this group of studies in relation to our understanding of science concept formation in the early years.

The Process of Science Concept Formation (Development over Time): Birth to Three

Representing the largest area of research examining science concept formation in the infancy period specifically, six of the seven studies identified in the review explored "how" children (as infants) are forming science concepts; the process of concept formation. Adopting a Cultural- Historical approach, Sikder and Fleer [66–68], examined infant-toddler's development of science concept formation within the family context. Based on Vygotsky's theory of concept formation, Sikder and Fleer [67] identified and categorised what they termed "small science", the toddlers' engagement and narration accompanying the small moments of their "everyday activities". They propose that "small science" can help explain how the everyday experiences of young children lay the foundation for the development of concrete "scientific" concepts. In a later paper, Sikder and Fleer [68] examined how these small science concepts become "ideal forms" from "real forms" in everyday life during the infant–toddler age period; findings indicated a conscious collaboration between parents and infants was key for developing small science concepts from rudimentary to mature form.

Providing a similarly cultural-historical examination of science concept formation, a large programmatic study (Fleers Conceptual PlayLab (CPL) in Melbourne, Australia) examined science concept formation in infancy in the context of an educational setting [72]. Using a model of intentional teaching called a Conceptual PlayWorld (CPW) [56], as an Educational Experiment, Fragkiadaki et al. [72], examined how infants in play-based settings,

develop scientific understandings about their everyday world. Using visual methods and concepts from Cultural Historical Theory (CHT), 4 key elements for introducing science concepts in infants' everyday educational reality were identified: (a) making dialectic interrelations between the everyday concept and the science concept, (b) consistently using a science language, (c) using appropriate analogies, and (d) using early forms of a scientific method.

Conducted within a similar cultural-historical framework Larsson [65], explored preschool children's opportunities for learning about friction focusing on four children and their everyday experiences within a Swedish preschool setting. Using the visual method of "shadowing" (direct observation using a video camera with a focus on a particular person), Larsson [65] found that during everyday situations and during play, a range of activities occur that place children in contact with the phenomenon of friction (e.g., when they shuffle, slide, or push). Children encounter the phenomenon of friction in many everyday situations, without reacting or reflecting upon it [65]. Larsson [65] suggests that such findings highlight the extent with which everyday play situations can be used by teachers to become more knowledgeable about children's current understandings of the phenomenon and ways of directing their attention toward understanding of friction in a more explicit manner, such as by inspiring children to new forms of play where the phenomenon of friction is prominent.

In similarity to Larsson [65], Klaa and Ohman [63] examined how children, as infants, encounter the phenomena of friction. Focusing on the way in which infants form science concepts through their actions in nature, Klaa and Ohman [63] provide insight into how the process of "meaning making" can be examined in infancy in the context of science. According to Klaa and Ohman [63], the investigation of meaning making should not be restricted to science concepts; analysis must relate to encounters with nature as one aspect among several in young children's lives. Accordingly, Klaa and Ohman [63] present and illustrate an approach that allows for the analysis of toddlers' meaning making when they physically encounter and experience nature in everyday practice. Based on Dewey's philosophy of Pragmatism, Klaa and Ohman [63] argue that their 'pragmatic action-oriented' perspective of learning can facilitate the analysis of toddlers' meaning making processes in early childhood education settings, contributing to a deeper understanding of the basic foundation for children's meaning making of nature.

Overall, there exists a plethora of methodological challenges associated with research involving children in the infancy period. The studies discussed provide valuable insight into ways in which the complexity of young children's science learning has and can be captured. The use of visual methodologies and Practical Epistemological Analysis, to gain insight into the child's everyday context were highlighted in the review and offer promising avenues for future exploration. In addition, this group of studies highlight the extent with which research into science concept formation in infancy specifically, has centred around examining the "process" of concept formation from a predominantly socio-cultural/cultural historical perspective. In contrast, the literature on older preschool age children has tended to adopt a constructivist approach which traditionally, has examined a child's demonstrated understanding of a particular science concept at a particular point in time; the "product" of concept formation. This difference is perhaps reflective of methodological challenges inherent in research with very young children.

## 4. Conclusions

Within the context of early childhood science education, we have a strong, empirically based understanding of science experiences for children aged three to six years. In contrast, our understanding of science learning as it occurs for children from birth to three, is extremely limited. We know very little about how and in what way children as infants and toddlers form conceptual understandings in science. Seeking to gain insight into current empirical understandings of science concept formation in the infancy–toddlerhood period, a review of the literature on early childhood science education (birth to six years) was

conducted. The literature review focused specifically (and for the first time) on children in the birth to three period.

Findings of the review illustrate a clear gap in the literature regarding science concept formation in the birth to three period; very little is known about infant and toddlers' thinking in science when considered in the context of birth to six years. Infants can and do engage in science [7] and the assumption that a particular developmental level must be reached before children can be taught science concepts has been challenged [72]. However, only a handful of empirical papers have examined science concept formation in the infancy–toddlerhood period specifically; we do not know enough about how science and scientific thinking can and does begin from birth [72]. The eminent need for empirical research to address this gap is highlighted by this review.

The lack of empirical research examining science concept formation in the birth to three period, identified in this review, can in part, be attributed to the methodological challenges inherent in research involving very young children. Determining how for example, the thinking of an infant with limited verbal skills can be documented, presents the researcher with a profound challenge. The fact that historically, infancy has been viewed as a period of helplessness may also explain the lack of empirical research identified; infants traditionally, have not been conceptualized as active, capable, "scientists". That infancy is a period in which there is no formal schooling, further complicates the construction of an empirical basis.

**5. Science Concept Formation in the Birth to Three Period: Key Points from the Literature**

In completing the review, a number of key points regarding the empirical literature on science concept formation in infancy and toddlerhood, were identified.

1.  Studies examining science concept formation in the birth to three period specifically, have focused primarily on exploring the process of concept formation; 6 of the 7 studies identified examined how young children develop their understandings of science concepts in an everyday context. This is in contrast to the literature on older pre-school age children (three to six years), where the tendency of research has been to focus on the relation between science concept formation and pedagogical practices; 24 of 50 studies (examining science concept formation in children aged three to six years) examined the relationships between teaching practices and children's conceptual development.

2.  Studies examining science concept formation in birth to three period, have tended to draw upon socio-cultural/cultural historical theory; 5 of the 7 studies identified, adopted a SC/CH theoretical framework. From a SC/CH perspective, cognitive development is conceptualised as a process whereby people move "through" understanding as opposed to towards it [86]. Concept formation is therefore, conceptualised as a dynamic process that must be examined as it occurs within and across differing contexts [54]. Research examining science learning in the birth to three period specifically, has focused on the "process" of concept formation. Within the broader Early Years Science Education Research (EYSER) literature, the tendency of research, to adopt a constructivist approach has been highlighted [6]. In contrast to SC/CH research, research adopting a constructivist approach has historically, examined children's demonstrated understandings of a particular science concept at a particular point in time. Thus, in contrast to the literature on science concept formation in infancy–toddlerhood, the literature within EYSER in general, has tended to examine the "product" of concept formation.

The greater emphasis placed on the process (as opposed product) of concept formation, seen in the literature on science in the infancy period, is, perhaps reflective of the difficulties inherent in using clinical methods to ascertain conceptual understandings of infants and toddlers (seen in broader constructivist approaches). Traditionally, EYSER research adopting a constructivist approach, has examined children's mental representations and understanding of science phenomena [61], findings are therefore based on children's

elicited responses, expressions of their thinking. Such methods, however, rely heavily on a certain level of cognitive awareness/verbal skills. The inherent challenges this presents when research involves very young children is highlighted.

In summary, the overall findings of the review provide clear evidence that a gap exists in the literature regarding infant and toddlers' thinking in science, very little is known when considered in the context of birth to six years. Urgent research attention is needed to take forward and to provide education systems with more evidence of infant and toddler thinking in science and what kinds of pedagogical approaches can amplify science thinking in the first period of children's lives. Without research within this early period, we do not have a sense of the continuum of science learning, under what conditions, what kind of methods are needed to study this period, or indeed how teachers can plan for infant-toddler development of science concept formation. This review paper gives the possibility to take stock of the gaps, and to point to new directions in science education research.

**Author Contributions:** The scope, focus and refinement of the review was conceptualised by M.F., G.F. and P.R., with modeling and mentoring of G.O., who synthesized and prepared tables and accompanying text. Final directions of the narrative and conclusion were jointly undertaken by all authors, but prepared by G.O. All authors have read and agreed to the published version of the manuscript.

**Funding:** This research was funded by the Australian Research Council [FL180100161].

**Institutional Review Board Statement:** Not applicable.

**Informed Consent Statement:** Not applicable.

**Data Availability Statement:** No new data were created or analyzed in this study. Data sharing is not applicable to this article.

**Acknowledgments:** We would like to acknowledge the research assistance of Sue March and the support of the Conceptual PlayLab PhD community at Monash University.

**Conflicts of Interest:** The authors declare no conflict of interest. The funders had no role in the design of the study; in the collection, analyses, or interpretation of data; in the writing of the manuscript, or in the decision to publish the results.

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
