# Peer review of "Early Childhood Science Education from 0 to 6: A Literature Review"

_education, doi:10.3390/educsci11040178_

Round 1

Reviewer 1 Report

The article provides an extensive overview of the literature on early childhood science education concerning issues as pedagogical practices, children’s conceptual understanding and the process of concepts formation. What makes the article very interesting is that it implicitly raises the issue of developing children’s scientific thinking in the birth to three years in the correlation to the educational framework. A particular challenge is the adoption of proper teaching and research methods for that period. Furthermore, the article is coherent and has a clear interconnection between the sub-sections.  

p.1(17). Science concepts instead of science concept?

p.3 I think that category 1A could contain the category 1B. What are the criteria which completely separate these two categories? The same for 3B and 1A or 1B.

p.3 Please provide clear definitions separating the category 2A from 3A.

p.3 Conceptual change in my opinion is related to the types of changes occurring in the content and organization of concepts but also to epistemological beliefs and in this sense differs from conceptual understanding which may have a limited character. Of course, learning in general is linked to change. If the authors connect in some way these two concepts it would be extremely fruitful to provide some argumentation about that.

p.3 (114-115) 2C: “Studies examining individual differences in science learning/conceptual understandings of pre-school age children”. “Differences” seems ambiguous here.  

p.3(120-121) 3A refers to learning processes adopted by the children. Is this also an issue of pedagogical practices (category 1) in the sense that a specific teaching method emerges or activates specific learning processes?  

p.3 (122-123). It should be explained why subcategory 3B was included in category 3 and not in category 1. To my sense the role of the teacher in creating conditions for children developing conceptual understanding can be perceived as an issue of pedagogical practices.

p.3 What is presented as category 1A in page 3, appears as 1B in pages 4 and 5.

p.5 (150) nam....names?

p.6 How did the authors deal with the problem of the researches that can be inserted into more than one category? For example, the study of Hadzigeorgiou (2002) has been entered in the category 1A(B) as it refers to pedagogical practices and specifically to hands–on activities, but Ι think that belongs also in 2A or 3A. The authors in the methodology section should indicate how the categories are appreciably distinguished from each other. I suppose it was authors’ methodological choice to enter each article into only one category evaluating the salient feature of each article. In any case the procedure the authors followed should be mentioned.

Author Response

We would like to thank the reviewer for highlighting the significance of the paper to the research of early childhood science education and pedagogical practices as well as the coherent way the paper is written. The comments of the reviewer were accurate and guide us in improving the clarity and quality of the manuscript.

Comment 1. p.1(17). Science concepts instead of science concept?

Response: We have addressed the comment. The change is marked in the manuscript – p.1 (17).

Comment 2. p.3 I think that category 1A could contain the category 1B. What are the criteria which completely separate these two categories? The same for 3B and 1A or 1B.

Response: We appreciate the reviewer’s comments and recognize that the differentiation between categories needs to be more clearly explained. In agreement with the reviewer we have now merged Category 1B into Category 1A.  The criteria separating 3B from 1A are that studies in Group 3B explore the way in which elements of the teachers’ role influence the process of concept formation (how early childhood teachers create the conditions for the formation of science concepts). These studies are therefore be differentiated from studies in Category 1 (The Teaching of Science Concepts) where the focus of research is on examining (specifically) the efficacy of specific teaching methods, interventions or instructional strategies in promoting scientific understanding. An explanation of the criteria differentiation has been added to the manuscript- p3 (131)

Comment 3. p.3 Please provide clear definitions separating the category 2A from 3A.

Response: The criteria separating 2A from 3A are that Studies in Group 3A examine the process of science concept formation (how children are developing an understanding of science concepts over time). These studies are therefore differentiated from those in Group 2A where the focus of research is on identifying children’s conceptual understandings and/ or ability to engage in science learning at a specific point in time. An explanation of the criteria differentiation has been added to the manuscript- p3 (126).

Comment 4.  p.3 Conceptual change in my opinion is related to the types of changes occurring in the content and organization of concepts but also to epistemological beliefs and in this sense differs from conceptual understanding which may have a limited character. Of course, learning in general is linked to change. If the authors connect in some way these two concepts it would be extremely fruitful to provide some argumentation about that.

Response: We thank the reviewer for this comment. We have now amended the manuscript changing the category name from ‘Conceptual Understandings/Change’ to ‘The Product of Concept Formation (Demonstrated Conceptual Understandings/ Capabilities) - p3 (1.08).

Comment 5. p.3 (114-115) 2C: “Studies examining individual differences in science learning/conceptual understandings of pre-school age children”. “Differences” seems ambiguous here.  

Response: We thank the reviewer for highlighting the ambiguity of this category label. We have now removed this group of studies from the review; studies in Group 2C and any related discussion have been removed from the manuscript. 

Comment 6. p.3(120-121) 3A refers to learning processes adopted by the children. Is this also an issue of pedagogical practices (category 1) in the sense that a specific teaching method emerges or activates specific learning processes?

Response: We thank the reviewer for this comment and although yes, some of the studies in Group 3A are an issue of pedagogical practice, they differ to studies in Category 1 in that the focus of studies in Category 1 is to examine (specifically) the efficacy of teaching practices/ interventions. In contrast, the focus of studies in 3A is to examine the process of science concept formation (how children are developing an understanding of science concepts over time). A more in-depth explanation of individual category criteria has been added to the manuscript- p.3 (126)).

Comment 7. p.3 (122-123). It should be explained why subcategory 3B was included in category 3 and not in category 1. To my sense the role of the teacher in creating conditions for children developing conceptual understanding can be perceived as an issue of pedagogical practices.

Response: We thank the reviewer for this comment. The criteria separating 3B from 1A are that studies in Group 3B explore the way in which elements of the teachers’ role, influence the process of concept formation (how early childhood teachers create the conditions for the formation of science concepts). These studies therefor differ to studies in Category 1 (pedagogical practices) where the focus of research is on examining the efficacy of specific teaching methods, interventions or instructional strategies in promoting scientific understanding. An explanation of the criteria differentiation has been added to the manuscript- p.3 (131).

Comment 8. p.3 What is presented as category 1A in page 3, appears as 1B in pages 4 and 5.

Response: We thank the reviewer for highlighting this and have amended the manuscript to address the comment- p.4

Comment 9. p.5 (150) nam....names?

Response: We thank the reviewer for highlighting this and have amended the manuscript to address the comment (see page 5, 162). 

Comment 10. p.6 How did the authors deal with the problem of the researches that can be inserted into more than one category? For example, the study of Hadzigeorgiou (2002) has been entered in the category 1A(B) as it refers to pedagogical practices and specifically to hands–on activities, but Ι think that belongs also in 2A or 3A. The authors in the methodology section should indicate how the categories are appreciably distinguished from each other. I suppose it was authors’ methodological choice to enter each article into only one category evaluating the salient feature of each article. In any case the procedure the authors followed should be mentioned.

Response: We thank the reviewer for this comment. We agree that a number of the studies could be inserted into more than category however the specific categorization system adopted, grouped studies in accordance with the overall essence of the paper (the overall research focus/ study aim). A more detailed explanation of category differentiation criteria has been added to the manuscript methodology section- p.3 (126).

Reviewer 2 Report

This paper presents the findings of the literature review, examining the scientific concept formation in the early years setting from 1990 to date. The scientific concept formation of children from zero to six years old, is examined by analysing sixty reviewed articles published since 1990 on this topic. 
The analysis methodology is rigorous and appropriate. The data obtained strongly support the conclusions and the discussion at work. 
My comments are two:
1) Various papers placed in any of the three categories developed by the authors, do not mention which scientific concepts they intend to teach. On the contrary, they use the term science in a generic way, and this is not a scientific concept, because it encompasses a large number of them. This fact could be commented, because the scientific concepts formation is related to specific knowledge.
2) The text concludes the non-existence of papers about the scientific concepts formation between zero and three years old, without discussing what this could be due to. 
I suppose that as the works of the Neopiagetian Karmiloff-Smith (1994) and other neuroscientifics as Stanislas Dehaene show, the methodological challenges of studying the learning of babies between 0 and 18 months are considerable, and also until three years old. Furthermore, it is a period in which there is no formal schooling, which also complicates the construction of an empirical basis. In any case, I leave the inclusion or not of these comments at the discretion of the authors.

Author Response

We would like to thank the reviewer for noting the rigorous and appropriate analysis methodology and the extent with which the data supports the conclusions and discussions. The comments of the reviewer were accurate and guide us in improving the clarity and quality of the manuscript.

Comment 1. Various papers placed in any of the three categories developed by the authors, do not mention which scientific concepts they intend to teach. On the contrary, they use the term science in a generic way, and this is not a scientific concept, because it encompasses a large number of them. This fact could be commented, because the scientific concepts formation is related to specific knowledge.

Response: We thank the reviewer for this valuable comment. We have now amended the manuscript to include the science concept(s) involved in each of the studies included in the review (see Tables 3,4 and 5).

Comment 2. The text concludes the non-existence of papers about the scientific concepts formation between zero and three years old, without discussing what this could be due to. I suppose that as the works of the Neopiagetian Karmiloff-Smith (1994) and other neuroscientifics as Stanislas Dehaene show, the methodological challenges of studying the learning of babies between 0 and 18 months are considerable, and also until three years old. Furthermore, it is a period in which there is no formal schooling, which also complicates the construction of an empirical basis. In any case, I leave the inclusion or not of these comments at the discretion of the authors.

Response: We thank the reviewer for this very valuable comment. We have now included in the manuscript suggested reasoning behind the lack of empirical research into science concept formation in infancy identified in the review- p.29 (492).